EMBO
Molecular Medicine

# VEGFA activates an epigenetic pathway upregulating ovarian cancer-initiating cells

Kibeom Jang[1,2], Minsoon Kim[1,2], Candace A Gilbert[1], Fiona Simpkins[1,3,†], Tan A Ince[1,4,5] (iD) & Joyce M Slingerland[1,2,6,*] (iD)

## Abstract

The angiogenic factor, VEGFA, is a therapeutic target in ovarian cancer (OVCA). VEGFA can also stimulate stem-like cells in certain cancers, but mechanisms thereof are poorly understood. Here, we show that VEGFA mediates stem cell actions in primary human OVCA culture and OVCA lines via VEGFR2-dependent Src activation to upregulate Bmi1, tumor spheres, and ALDH1 activity. The VEGFA-mediated increase in spheres was abrogated by Src inhibition or *SRC* knockdown. VEGFA stimulated sphere formation only in the ALDH1[+] subpopulation and increased OVCA-initiating cells and tumor formation *in vivo* through Bmi1. In contrast to its action in hemopoietic malignancies, DNA methyl transferase 3A (DNMT3A) appears to play a pro-oncogenic role in ovarian cancer. VEGFA-driven Src increased DNMT3A leading to *miR-128-2* methylation and upregulation of Bmi1 to increase stem-like cells. *SRC* knockdown was rescued by antagomir to miR-128. *DNMT3A* knockdown prevented VEGFA-driven *miR-128-2* loss, and the increase in Bmi1 and tumor spheres. Analysis of over 1,300 primary human OVCAs revealed an aggressive subset in which high VEGFA is associated with *miR-128-2* loss. Thus, VEGFA stimulates OVCA stem-like cells through Src-DNMT3A-driven *miR-128-2* methylation and Bmi1 upregulation.

**Keywords** Bmi1; cancer stem-like cell; epigenetic; miR-128; VEGFA
**Subject Categories** Cancer; Chromatin, Epigenetics, Genomics & Functional Genomics; Urogenital System

## Introduction

Developing cancers rapidly outstrip the diffusion capacity of nutrients and oxygen and must form new blood vessels via angiogenesis (Bergers & Benjamin, 2003). They must also maintain self-renewal capacity despite adverse conditions of suboptimal pH, nutrient, and oxygen availability. Vascular endothelial growth factor A, VEGFA, is a key angiogenic factor that alters the endothelial cell niche to promote new vessel formation (Bergers & Benjamin, 2003). In various cancers, VEGFA stimulates not only angiogenesis but also tumor growth, metastasis, and survival (Wu *et al*, 2006; Hu *et al*, 2007; Paez-Ribes *et al*, 2009).

The importance of VEGFA in angiogenesis and its frequent upregulation in human cancers (Goel & Mercurio, 2013) stimulated development of VEGF- and VEGF receptor-targeted therapies. The monoclonal antibody, bevacizumab, blocks VEGFA interaction with receptors VEGFR1 and 2 (Ferrara, 2004). Despite initial promise, VEGF-targeted therapies have shown limited efficacy, with short responses in most solid tumors (Eskander & Tewari, 2014). Hypoxia resulting from inhibition of angiogenesis upregulates VEGFA expression, contributing to aggressive disease recurrence and angiogenic therapy failure (Ebos *et al*, 2009; Paez-Ribes *et al*, 2009). In ovarian cancer, bevacizumab significantly increases progression-free survival (PFS) compared with chemotherapy alone in advanced disease, and more recently VEGFR-targeting agents were shown to significantly increase PFS (Eskander & Tewari, 2014) and overall survival (OS; Witteveen *et al*, 2013). Bevacizumab is approved for use with non-platinum chemotherapy in Europe and the USA for platinum-resistant ovarian cancer; however, reponses are of short duration and resistance rapidly emerges (Eskander & Tewari, 2014).

Increasing evidence indicates that cancer stem-like cells (CSCs) comprise a distinct self-renewing subpopulation that can generate cancerous progeny with reduced replicative potency (Dalerba *et al*, 2007). CSCs are important therapeutic targets: they may not only initiate tumors but also mediate recurrence and metastasis (Magee *et al*, 2012). Most anticancer drugs kill the bulk cancer population.

1   Braman Family Breast Cancer Institute at Sylvester Comprehensive Cancer Center, University of Miami Miller School of Medicine, Miami, FL, USA
2   Department of Biochemistry & Molecular Biology, University of Miami Miller School of Medicine, Miami, FL, USA
3   Department of Obstetrics & Gynecology, University of Miami Miller School of Medicine, Miami, FL, USA
4   Department of Pathology, University of Miami Miller School of Medicine, Miami, FL, USA
5   Interdisciplinary Stem Cell Institute, University of Miami Miller School of Medicine, Miami, FL, USA
6   Department of Medicine, University of Miami Miller School of Medicine, Miami, FL, USA
    *Corresponding author. Tel: +1 305 243 7265; Fax: +1 305 243 6170; E-mail: jslingerland@med.miami.edu
    †Present address: Department of Obstetrics & Gynecology, University of Pennsylvania, Philadelphia, PA, USA

   

CSCs either proliferate too slowly for targeting by cycle active drugs, or escape chemotherapy by drug expulsion or greater DNA repair, leading to recurrence (Magee *et al*, 2012). CSCs share properties with normal tissue stem cells, expressing discrete surface markers and forming spheres when seeded at single cell density (Visvader & Lindeman, 2012). Subpopulations of many malignancies, including OVCA, with aldehyde dehydrogenase 1 activity (ALDH1[+]) are enriched for CSC properties *in vitro* and the ability to initiate tumors in immunocompromised mice (Ginestier *et al*, 2007; Landen *et al*, 2010). Several surface markers have been proposed to characterize OVCA stem-like cells (Shah & Landen, 2014). The ALDH1[+] subpopulation of primary ovarian cancers (Stewart *et al*, 2011) and in OVCA lines (Silva *et al*, 2011; Shah & Landen, 2014) is enriched in tumor-initiating cells *in vivo* and by prior chemotherapy exposure (Landen *et al*, 2010). While certain cytokines increase CSC and enhance tumor initiation *in vivo* (Zhao *et al*, 2014), extracellular growth factors and signaling pathways that stimulate CSC expansion are poorly characterized. A greater understanding of pathways governing CSC may permit the design of more effective anticancer treatments.

VEGFA is not only a potent angiogenic factor, it also stimulates stem-like cells in both normal and cancer tissues. VEGFA maintains normal stem cell populations in hemopoietic (Gerber *et al*, 2002), endothelial (Kane *et al*, 2011), and neuronal tissues (Calvo *et al*, 2011). VEGFA was recently shown to increase tumor-initiating stem-like cells in skin and breast cancers (Beck *et al*, 2011; Goel *et al*, 2013; Zhao *et al*, 2014). Pathways activated by VEGFA that increase cancer stem-like cells and tumor initiation are largely uncharacterized. Since VEGFA is frequently overexpressed in OVCA (Yu *et al*, 2013) and VEGF/VEGFR-targeted therapies have significant activity in this cancer (Eskander & Tewari, 2014), we investigated whether VEGFA drives ovarian CSC expansion and sought to identify targetable pathways mediating these effects.

MicroRNAs (miRNAs) are increasingly implicated in CSC regulation (Takahashi *et al*, 2014). These small non-coding RNAs bind the 3′ untranslated region (3′ UTR) of target genes to inhibit gene expression. miRNAs regulate targets essential for normal and malignant stem-like cell self-renewal (Takahashi *et al*, 2014) and are often misregulated in cancer (Croce,2009). Oncogenic miRNAs target tumor suppressors and increase drug resistance and metastasis. In contrast, tumor suppressor miRNAs are frequently downregulated in cancer (Croce, 2009). Here, we investigated the role of VEGFA as a driver of stem-like cell expansion in OVCA. This work reveals a novel pathway linking VEGFA to miRNA-dependent CSC regulation. We show that VEGFA activates Src and induces *DNMT3A* to methylate *miR-128-2*, leading to increased Bmi1 and OVCA stem-like cell expansion.

# Results

## VEGFA increases sphere formation and ALDH1 activity in OVCA populations

While VEGFA is an angiogenic agent and therapeutic target in OVCA, the possibility that the limited effects of VEGF-targeted therapies and emergence of resistance might be due, in part, to VEGFA effects on ovarian CSC has not been evaluated. To investigate whether VEGFA stimulates ovarian CSCs, we used three models. Since > 60% of OVCAs express the estrogen receptor α (ER), we used the well-established ER[+] line, PEO1R, derived from human high-grade serous OVCA ascites (Langdon *et al*, 1994). Results were validated using an ER- high-grade serous human OVCAR8 line (Slayton, 1984; Domcke *et al*, 2013). Since OVCA lines diverge from primary tumors over time (Ince *et al*, 2015) and since OVCA may represent different cancers with different molecular origins, results were validated using early-passage OCI-C5X, a primary OVCA culture derived from a clear cell OVCA. OCI-C5X faithfully represents the molecular and cellular phenotype of the original patient's clear cell tumor and is one of 25 new ovarian cancer cultures established by Ince by immediate culture of primary cancer in Ovarian Carcinoma Modified Ince medium, OCMI (Ince *et al*, 2015).

Sphere formation from a single CSC seeded in low adhesion conditions is a measure of stem-like cell abundance *in vitro* (Visvader & Lindeman, 2012). Prior work showed VEGFA and a network of pro-inflammatory cytokines increase breast CSC abundance, but required prolonged exposure for full effect (Zhao *et al*, 2014; Picon-Ruiz *et al*, 2016). VEGFA (10 ng/ml) effects were assayed over short- and long-term exposures (1, 3, and 7 days). A 7-day exposure, but not shorter intervals, significantly increased sphere formation by PEO1R, OVCAR8, and OCI-C5X cells seeded without further VEGFA. All sphere assays were carried out in limiting dilutions and sphere formation could not be accounted for by aggregation. VEGFA-blocking antibody, bevacizumab, and 2C3 antibody that blocks VEGFR2 both inhibited VEGFA-stimulated sphere formation (Fig 1A). Neither antibody alone decreased baseline sphere formation, suggesting VEGFA does not drive basal CSC self-renewal, but augments CSC recruitment in these OVCA models. VEGFA was not a mitogen in these models. Cell cycle profiles were not changed by VEGFA exposure for 48 h in 2D and were not affected by 7 days in sphere culture (representative data from OVCAR8, Fig EV1A). Furthermore, the % apoptotic cells in PEO1R as assayed by annexin V staining was unchanged by VEGFA exposure over 7 days (Fig EV1B) and VEGFA did not increase cell numbers over time in unsorted cells (Fig EV1C).

The ALDH1[+] population of OVCA lines (Silva *et al*, 2011; Shah & Landen, 2014) and primary tumors (Stewart *et al*, 2011) is enriched for CSC *in vitro* and tumor-initiating cells *in vivo*. VEGFA exposure increased ALDH1[+] cell abundance in all three models, including early-passage OCI-C5X culture (Ince *et al*, 2015; Fig 1B). Thus, prolonged VEGFA exposure increases the abundance of sphere forming and ALDH1[+].

## VEGFA increases expression of the stem-like cell regulator Bmi1

Several embryonic stem cell transcription factors (ES-TFs) govern embryonic stem cells (ES) self-renewal, induce pluripotency in skin fibroblasts (Li *et al*, 2010), and have been implicated in CSC self-renewal in a number of different cancers (Zhang *et al*, 2008). B cell-specific Moloney murine leukemia virus integration site 1 (Bmi1) is part of the polycomb-repressive complex 1 (PRC1) that regulates chromatin remodeling during development (Siddique & Saleem, 2012). Bmi1 promotes normal hematopoietic and neural stem cell expansion (Park *et al*, 2003; Molofsky *et al*, 2005) and can upregulate malignant stem-like cells in part through changes in ES-TFs (Lessard & Sauvageau, 2003; Siddique & Saleem, 2012). Bmi1 has

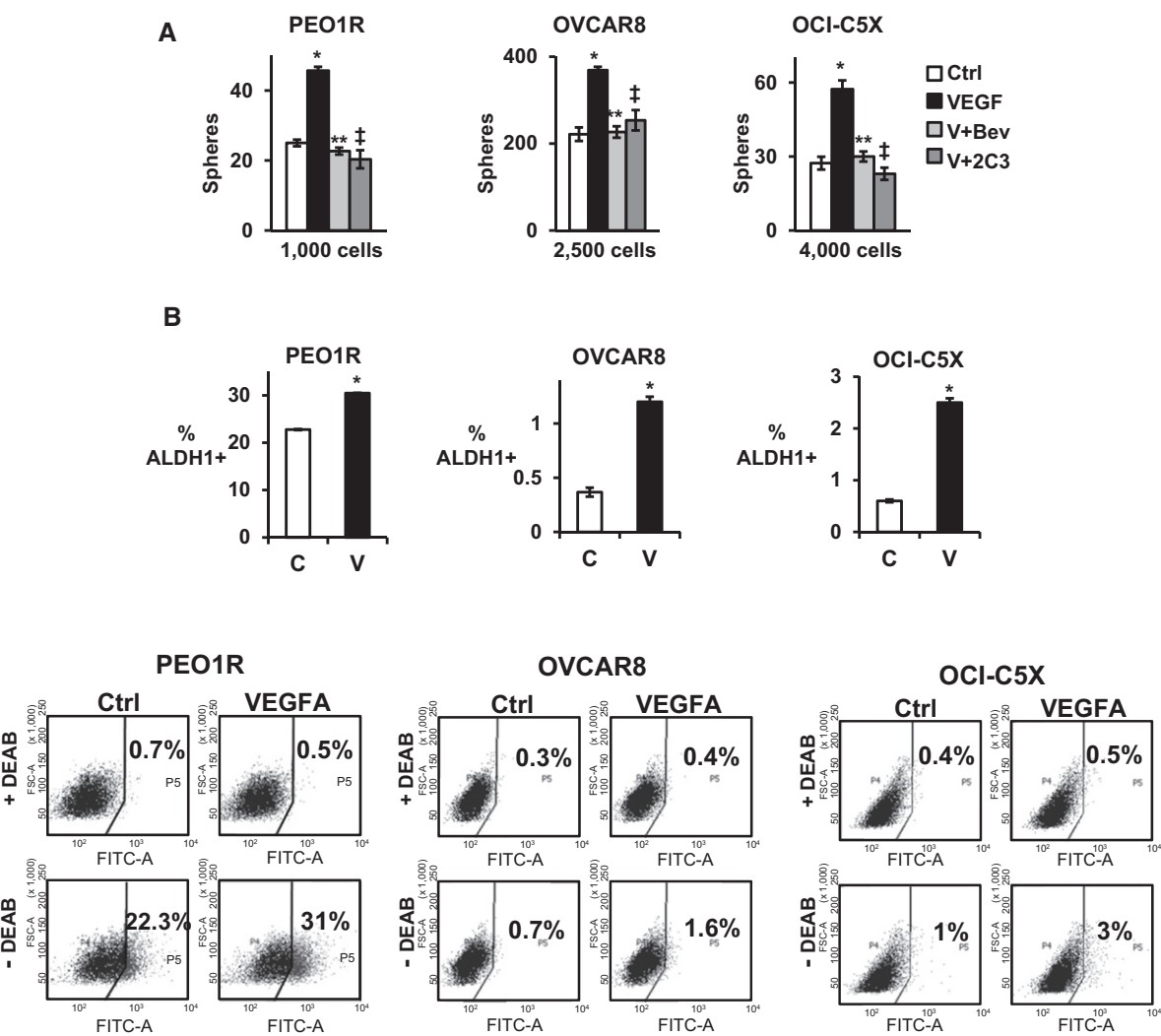

**Figure 1.  VEGFA increases sphere-forming and ALDH1-positive OVCA cells.**

See also Fig EV1.

A   Indicated cells were pre-treated for 7 days with either 10 ng/ml VEGFA, VEGFA + 50 μg/ml bevacizumab, VEGFA + 15 μg/ml 2C3, or no cytokine control and plated into sphere assays. Spheres > 75 μm were counted at 14 days for PEO1R and at 21 days for OVCAR8 and OCI-C5X. All assays were performed in triplicate biologic repeats with at least three technical repeats within each assay and graphed data represent mean ± SEM. Differences between multiple treatment groups were compared by ANOVA. PEO1R; *$P$ = 0.0003, **$P$ = 0.23, ‡$P$ = 0.24, OVCAR8; *$P$ = 0.0022, **$P$ = 0.85, ‡$P$ = 0.41, OCI-C5X; *$P$ = 0.0051, **$P$ = 0.55, ‡$P$ = 0.38.

B   Cells were treated with ± VEGFA for 7 days and the proportion of ALDEFLUOR positive (% ALDH1[+]) assayed by flow cytometry and mean ± SEM graphed for repeat assays. ALDH1[+] cells in controls versus VEGFA-treated cells were compared by Student's $t$-test. PEO1R; *$P$ = 0.000003, OVCAR8; *$P$ = 0.0033, OCI-C5X; *$P$ = 0.00048.

been shown to be upregulated in cancers, including OVCA (Siddique & Saleem, 2012), but its role in ovarian CSC expansion has not been established.

To assay their potential involvement in VEGFA-stimulated ALDH1 activity and sphere formation, we tested VEGFA effects on Bmi1 and ES-TFs in PEO1R, OVCAR8, and OCI-C5X. Basal Bmi1 levels were increased by 6 h and remained elevated at 7 days (Fig 2A, left). Densitometry of Western blots on repeat assays showed Bmi1 protein increased by 2.0 ± 0.02 fold in PEO1R, 4.7 ± 0.1 fold in OVCAR8, and 1.9 ± 0.09 fold in OCI-C5X after 7 days of VEGF exposure. Notably, while VEGFA also increased ES-TF levels, including cMyc (by 3.5 ± 0.1 fold by day 1), Oct4 (2.2 ± 0.06 fold, day 2), and Klf4 (4.1 ± 0.3 fold day 2; Fig EV2),

these rose after the increase in Bmi1. Thus, we assayed the role of Bmi1 in VEGFA-mediated ovarian CSC effects.

## Increased sphere formation after VEGFA exposure is Src- and Bmi1-dependent

Src is frequently overexpressed in human OVCA (Simpkins et al, 2012) and promotes tumor growth (Kim et al, 2009). Since Src was recently shown to mediate cytokine-driven CSC upregulation (Picon-Ruiz et al, 2016), we tested its potential as a mediator of effects of VEGFA on stem-like cells. VEGFA caused sustained Src activation with an increase in pSrc of 4.1 ± 0.42 fold in PEO1R, and 4.8 ± 0.08 fold in OVCAR8 and 1.8 ± 0.04 fold in OCI-C5X within

7 days (Fig 2B). Src inhibition by saracatinib (AZD0530) downregulated basal Bmi1 levels. Furthermore, saracatinib addition during the last 48 h of a 7-day VEGFA exposure prevented the increase in Bmi1 by VEGFA in all three ovarian models (Fig 2C). To test effects of Src inhibition on the sphere-forming population, cells were treated with AZD0530 in the last 48 h of a 7-day VEGFA exposure, followed by a 2-day washout to allow recovery of asynchronous cycling prior to seeding into sphere assays (Fig EV3A). Src inhibition followed by drug washout and prior *BMI1* siRNA knockdown (Fig 2D) each decreased sphere formation below that of controls and prevented the VEGFA-mediated increase in sphere formation in both lines, and in OCI-C5X primary culture (Fig 2E). This loss of sphere formation could not be attributed to changes in cell cycling or viability, since neither *BMI1* knockdown nor Src inhibition followed by washout-affected cell cycle profiles or viable cell numbers of cells prior to seeding (Fig EV3A and B). Thus, Src kinase action appears to govern basal Bmi1 expression and both are required for the VEGFA-mediated increase in sphere formation.

### VEGFA increases ovarian tumor-initiating cells via Bmi1 *in vivo*

Effects of VEGFA and Bmi1 on ovarian tumor-initiating cell abundance were further investigated *in vivo*. Limiting dilution tumor-initiating cell assays, injecting between 100 and 100,000 cells, showed that sustained VEGFA exposure over 7 days prior to injection increased PEO1R tumor-initiating cell abundance. *Ex vivo* exposure to VEGFA decreased tumor latency and more animals formed tumors from VEGFA-exposed cells than from cells without VEGFA pre-treatment. *BMI1* knockdown prevented the VEGFA-mediated increase in tumor-initiating cell abundance (Fig 3A). Note that VEGFA was not a mitogen in this model and did not affect apoptosis (Fig EV1). *BMI1* siRNA did not impair proliferation or viability (Fig EV3). The tumor-initiating cell frequency in VEGFA-exposed cells was 1/2,018, compared with 1/21,607 in non-VEGFA-exposed cells and 1/20,313 in VEGFA-exposed cells pre-treated with siRNA to *BMI1*, as calculated by L-Calc™ Limiting Dilution Software (Fig 3B). Thus, VEGFA increases tumor-initiating OVCA cell abundance *in vivo* and this is Bmi1 dependent.

### VEGFA repression of *miR-128-2* is Src dependent

Bmi1 is regulated by miR-128, a 21 nucleotide (ucacagugaaccggucucuuu) that targets the *BMI1* 3′ UTR (Godlewski *et al*, 2008; Zhu *et al*, 2011; Jin *et al*, 2014). Mature miR-128 is encoded by two miRs, *miR-128-1* and *miR-128-2*. qPCR with primers that distinguish pre-*miR-128-1* and pre-*miR-128-2* showed that VEGFA significantly reduced *miR-128-2* but not *miR-128-1* expression after 7 days (Fig EV4). Since pre-miRs are unstable and less abundant, subsequent work used primers to detect mature miR-128. VEGFA downregulated miR-128 in all OVCA models (Fig 4A). To test whether VEGFA relieves miR-128 targeting of the *BMI1* 3′ UTR, VEGFA-exposed and control cells were transfected with a *BMI1* 3′ UTR luciferase reporter. VEGFA increased *BMI1* 3′ UTR luciferase reporter activity in both OVCA lines and in OCI-C5X (Fig 4B). Thus, Bmi1 upregulation by VEGFA results from decreased inhibitory occupancy of the *BMI1* 3′ UTR.

To test whether Src activation drives VEGFA-mediated miR-128 loss, cells were treated with or without VEGFA, and with or without

saracatinib (AZD0530). Saracatinib increased basal miR-128 and abrogated miR-128 downregulation by VEGFA in all three models (Fig 4C). Effects of the Src family kinase inhibitor, saracatinib, were verified using siRNA-mediated *SRC* knockdown. Since saracatinib effectively inhibits several Src kinase family members, the requirement for Src was verified using *SRC* knockdown by siRNA with three different oligonucleotides. *SRC* knockdown increased basal *miR-128* expression, and *SRC* siRNA transfection 48 h prior to addition of VEGFA abrogated the VEGFA-driven loss of *miR-128* expression in both OVCA lines and in the primary OCI-C5X culture (representative data, Fig 4D). *SRC* knockdown not only increased basal miR-128, but it also prevented the VEGFA-driven increase in sphere formation. AntagomiR-128 rescued the inhibition of sphere formation by si*SRC* (Fig 4E), consistent with the notion that VEGFA-mediated action on miR-128 is downstream of Src. Thus, Src appears to govern basal miR-128 expression and Src activation is required for miR-128 repression following VEGFA exposure.

### VEGFA repression of miR-128 requires DNA methyltransferase activity

VEGFA exposure had effects long after VEGFA withdrawal to increase OVCA tumor initiation *in vivo* (Fig 3); thus, we speculated that VEGFA might regulate *miR-128-2* epigenetically. The extent to which miRNAs are epigenetically regulated is not fully understood (Suzuki *et al*, 2012). Tumor suppressor miRs are downregulated in cancers (Croce, 2009) through DNA methylation, since their repression is relieved by the methylation inhibitor 5′-azacytidine (Saito *et al*, 2006; Suzuki *et al*, 2012). While such miRNAs are methylation sensitive, the methyltransferases governing miRNA expression have not been fully characterized. 5′-azacytidine increased miR-128 expression in controls and blocked miR-128 downregulation by VEGFA in PEO1R and OCI-C5X (Fig 4F). Thus, VEGFA downregulation of miR-128 requires DNA methyltransferase action.

### VEGFA upregulates Bmi1 through Src-mediated DNMT3A induction and *miR-128-2* methylation

DNA methyltransferase (DNMT) 3A plays an important role to restrain gene expression and is deregulated in human malignancies (Fernandez *et al*, 2012). To test potential DNMT involvement in VEGFA-driven *miR-128-2* repression, and since DNA methyltransferases genes, *DNMT3A, DNMT3B, DNMT1*, are often coregulated and elevated in cancer stem-like cells (Yang *et al*, 2015), we assayed all three methyltransferases. VEGFA increased *DNMT3A* but not *DNMT1* and *DNMT3B* expression (Fig EV5A). Src inhibition decreased *DNMT3A* expression in untreated cells and prevented the increase in *DNMT3A* expression by VEGFA (Fig 5A, left). DNMT3A protein was increased by 2.6 ± 0.11 fold in PEO1R and by 1.6 ± 0.08 fold in OCI-C5X after 7-day VEGFA exposure (Fig 5A and B). Addition of saracatinib (AZD0530) abolished the VEGFA-stimulated DNMT3A increase and decreased basal DNMT3A protein levels (Fig 5A, right). Notably, siRNA-mediated *SRC* knockdown decreased basal DNMT3A and Bmi1 levels and prevented the VEGFA-driven upregulation of both DNMT3A and Bmi1 in PEO1R and OCI-C5X (Fig 5B).

*DNMT3A* knockdown modestly decreased baseline sphere formation and prevented the increase in sphere formation by

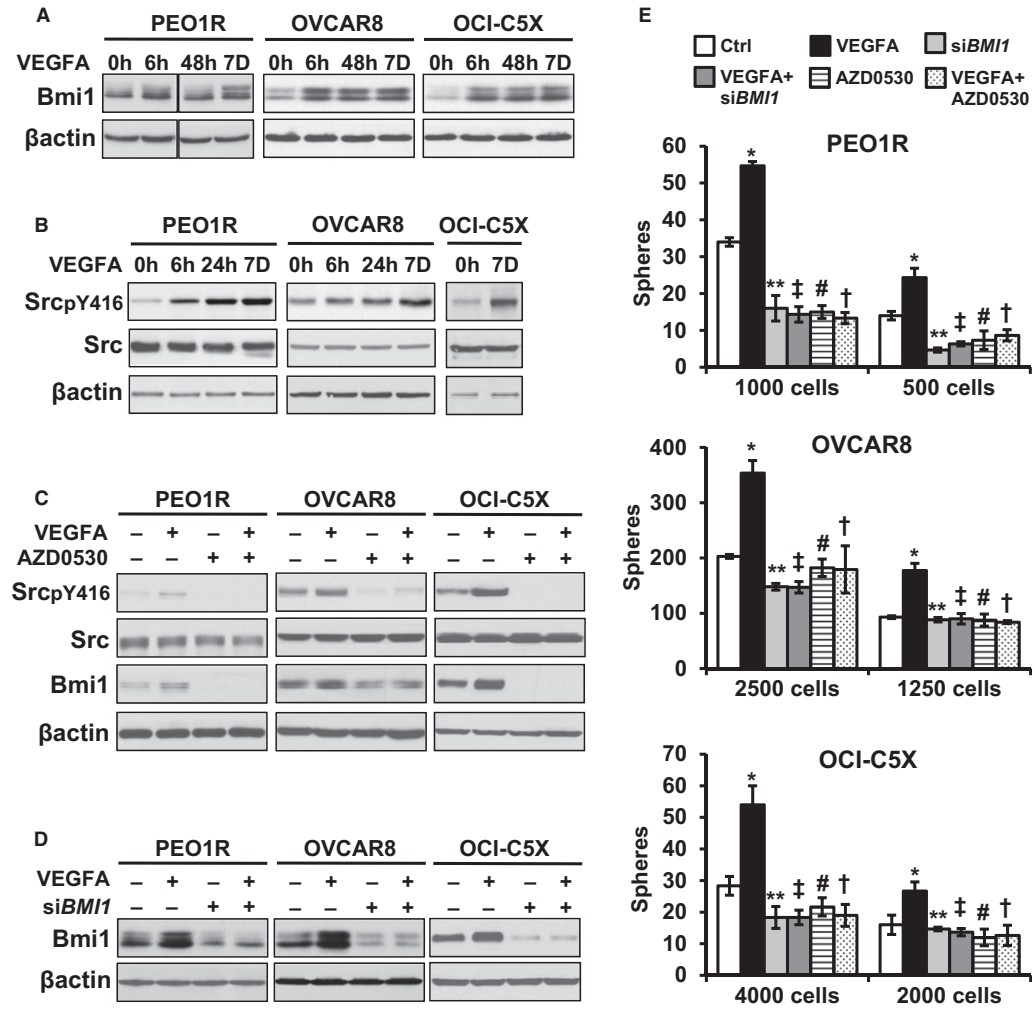

**Figure 2. VEGFA increases OVCA sphere formation via Src-mediated Bmi1 upregulation. VEGFA effects on indicated proteins.**

See also Figs EV2 and EV3.

A, B   Bmi1 (A), and total and Y416-phosphorylated Src (pSrc) (B), at times indicated.

C   Cells were treated for 7 days with or without VEGFA, ± Src inhibition by 1 μM saracatinib (AZD0530) during the last 48 h. Western blots show Src, pSrc, and Bmi1 levels.

D   Cells were transduced with siRNA BMI1 or scrambled controls 48 h prior to VEGFA treatment for 7 days and then recovered for Western blot.

E   Cells were transduced with either siBMI1 or control siRNA for 48 h prior to VEGFA addition for 7 days (± siBMI1) or treated with VEGFA for 7 days with or without Src inhibitor (AZD0530, 1 μM) during the last 48 h followed by a 2-day washout without drug or cytokine prior to plating of spheres into limiting dilution sphere formation. All graphed data show mean ± SEM for at least 3 different biologic experiments with at least three technical repeats within each assay. Differences between multiple treatment groups were compared by ANOVA. PEO1R; 1,000 cells: *$P < 0.0001$, **$P < 0.0001$, ‡$P < 0.0001$, #$P < 0.0001$, †$P < 0.0001$, 500 cells: *$P < 0.0001$, **$P = 0.0002$, ‡$P = 0.001$, #$P = 0.003$, †$P = 0.0143$, OVCAR8; 2,500 cells: *$P < 0.0001$, **$P = 0.0328$, ‡$P = 0.0297$, #$P = 0.6599$, †$P = 0.5492$, 1,250 cells: *$P = 0.0001$, **$P = 0.9344$, ‡$P = 0.9889$, #$P = 0.88954$, †$P = 0.5744$, OCI-C5X; 4,000 cells: *$P < 0.0001$, **$P = 0.0381$, ‡$P = 0.0381$, #$P = 0.2128$, †$P = 0.0545$, 2,000 cells: *$P = 0.0008$, **$P = 0.9399$, ‡$P = 0.6718$, #$P = 0.2289$, †$P = 0.3716$.

VEGFA (Fig 5C). DNMT3A was required for both VEGFA-driven loss of miR-128 and for the increase in Bmi1 protein (Fig 5D and E). Thus, VEGFA-driven miR-128 repression, Bmi1 upregulation, and stem-like cell expansion appear to be DNMT3A dependent.

*miRNA-128-2* DNA methylation was analyzed by genomic bisulfite sequencing in PEO1R and in early-passage OCI-C5X. The *miR-128-2* region contains ten CpG sites in a 347-bp sequence (nucleotide −166 to +181; Fig 5F). VEGFA increased the frequency of methylated CpGs in *miR-128-2* in both PEO1R and OCI-C5X (Figs 5G and EV5B) consistent with its downregulation of miR-128 expression.

Saracatinib (AZD0530) and 5′-azacytidine both decreased baseline methylation and prevented the VEGFA-mediated increase in methylated CpGs in *miR-128-2* (Figs 5G and EV5B). Notably, baseline miR-128 expression was lower and *miR-128-2* methylation was higher in PEO1R than in early-passage OCI-C5X (Figs 5G and EV5B and C), supporting the notion that *miR-128-2* expression is governed by DNA methylation. DNA methylation of *miR-128-2* was strongly inversely correlated with its expression in both PEO1R ($R^2 = -0.7513$) and in the primary OCI-C5X models ($R^2 = -0.9559$; Fig 5H).

A more detailed kinetic time course showed that short-term VEGFA exposure rapidly activates VEGFR2pY1175 leading to

## A

**Ctrl** — **VEGFA** — **siBMI1** — **VEGFA + siBMI1**

**100,000 cells**

% Tumor Free vs Weeks

**10,000 cells**

% Tumor Free vs Weeks

**1,000 cells**

% Tumor Free vs Weeks

**100 cells**

% Tumor Free vs Weeks

## B

| Group | Limiting Dilution Tumors / Implant | | | | T-ISC Freq. (1 in/..) | P value (vs ctrl) |
|---|---|---|---|---|---|---|
| | 100,000 cells | 10,000 cells | 1,000 cells | 100 cells | | |
| Control | 4 / 5 | 4 / 7 | 3 / 7 | 0 / 10 | 21,607 | - |
| VEGFA | 5 / 5 | 7 / 8 | 4 / 8 | 4 / 10 | 2,018 | 0.0005 |
| siBMI1 | 4 / 5 | 4 / 8 | 3 / 8 | 1 / 10 | 20,313 | 0.9291 |
| VEGFA+siBMI1 | 4 / 5 | 4 / 8 | 3 / 8 | 1 / 10 | 20,313 | 0.9291 |

**Figure 3.   The VEGFA-mediated increase in OVCA-initiating stem-like cell abundance *in vivo* is Bmi1 dependent.**
A   Tumor formation from limiting dilutions of inoculated cells (100,000, 10,000, 1,000, 100 cells) is graphed as % of tumor-free animals/time (weeks).
B   Tumor formation is tabulated and T-ISC frequency is calculated.

increased SrcpY416. Both DNMT3A and Bmi1 increased gradually within the first 6–12 h after VEGFA addition. miR-128 levels, in turn, fell significantly within the first 6–12 h, and the decline was sustained and progressive over the next 7 days (Fig 6A and B). Thus, while VEGFR2 and Src signaling is triggered within minutes, the signal increases progressively and the epigenetic consequences of Src-dependent DNMT3A upregulation and miR-128 loss were manifest progressively over the successive 3–4 DNA replication cycles over 7 days. Notably *VEGFA* gene expression was also increased over the prolonged time course, providing a feed-forward mechanism to further activate Src and maintain this process (Fig 6C). In keeping with an epigenetic effect and not merely a signaling-dependent mechanism, the increase in sphere-forming ability was not fully felt until after several cell divisions: VEGFA exposure for 48 h did not significantly increase sphere; a 7-day

exposure was required (Fig 6D). These data support that long-term VEGFA exposure is required to increase the abundance of CSC.

### VEGFA confers self-renewal of ALDH1+ cells

Our initial assays showed VEGFA increased the proportion of ALDH1+ cells. We next assayed Src activation, DNMT3A levels, and Bmi1 in cells with different ALDH1 activity. When ALDH1-positive and ALDH1-negative populations were isolated by flow cytometry, ALDH1+ PEO1R cells showed higher Bmi1 (5.1 ± 0.08 fold), Src activation (pSrc, 1.8 ± 0.02 fold), and DNMT3A levels (1.5 ± 0.07 fold) than ALDH1− cells, supporting the notion that Src and Bmi1 are drivers of ovarian CSCs in this model (Fig 7A). Furthermore, ALDH1+ population also expressed lower miR-128 than ALDH1− cells (Fig 7B).

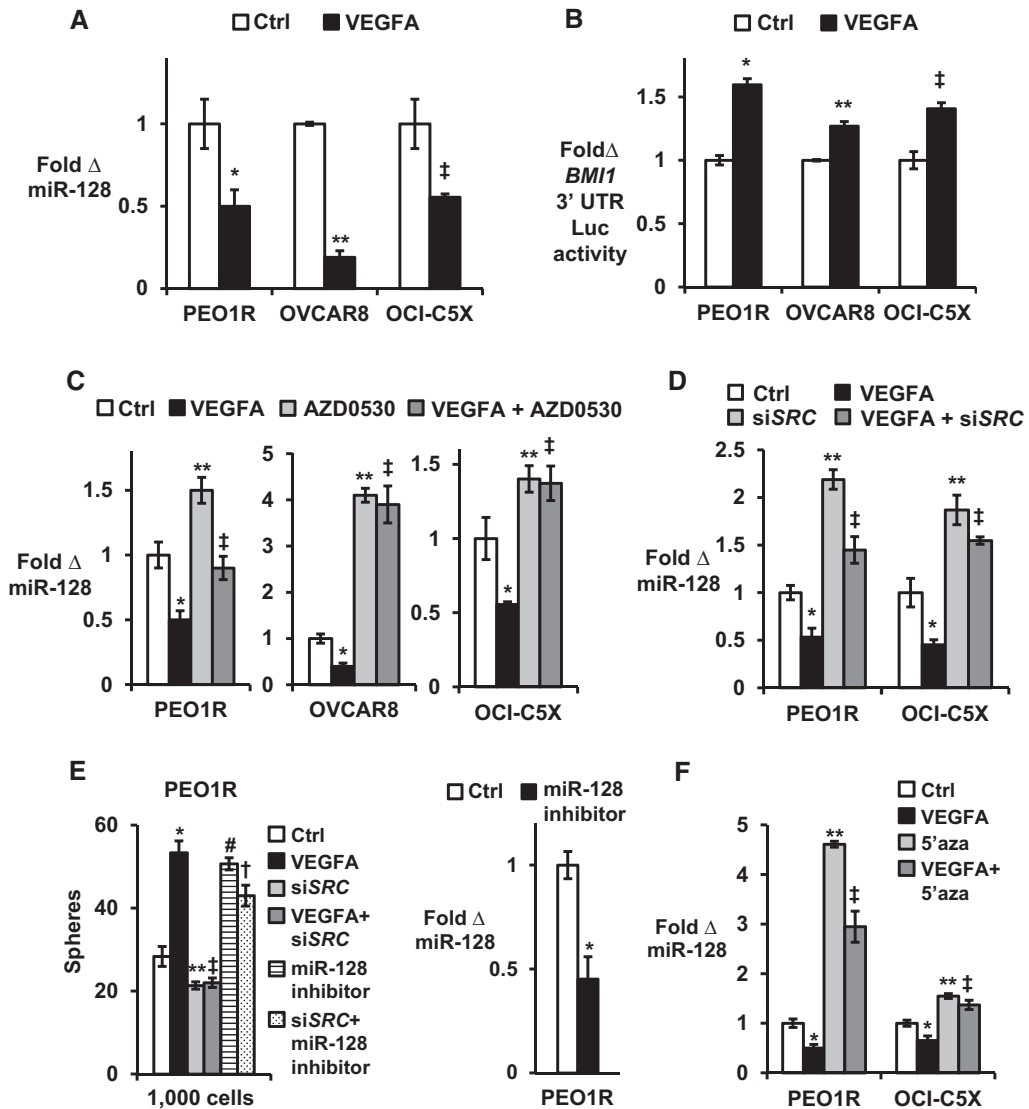

**Figure 4. VEGFA downregulates miR-128 expression in a Src-dependent manner.**

A  VEGFA treatment for 7 days decreases miR-128 expression as assayed by qPCR. *$P = 0.03$, **$P = 0.0000003$, ‡$P = 0.01$.

B  VEGFA increases *BMI1* 3′ UTR luciferase activity. *$P = 0.00015$, **$P = 0.00046$, ‡$P = 0.0022$.

C  Effects of VEGFA alone, of saracatinib (AZD0530, 1 μM) alone, or saracatinib added during the final 48 h of 7-day VEGFA treatment on miR-128 expression compared with mock-treated controls (Ctrl). PEO1R; *$P = 0.0005$, **$P = 0.0005$, ‡$P = 0.5522$, OVCAR8; *$P = 0.0359$, **$P < 0.0001$, ‡$P < 0.0001$, OCI-C5X; *$P = 0.0064$, **$P = 0.0113$, ‡$P = 0.017$.

D  Effects of VEGFA alone for 7 days, of siRNA to *SRC* (si*SRC*) alone, or si*SRC* transfection 48 h prior to addition VEGFA for 7 days on miR-128 expression compared with mock-treated controls (Ctrl). PEO1R; *$P = 0.019$, **$P = 0.0008$, ‡$P = 0.047$, OCI-C5X; *$P = 0.027$, **$P = 0.016$, ‡$P = 0.024$.

E  Cells were transduced with either si*SRC* or control siRNA for 48 h prior to VEGFA addition for 7 days (± si*SRC*), or treated with antagomiR-128 for 48 h with or without si*SRC* and plated into sphere assays. *$P = 0.0026$, **$P = 0.052$, ‡$P = 0.076$, #$P = 0.0014$, †$P = 0.014$. AntagomiR-128 decreases miR-128 expression as assayed by qPCR. *$P = 0.012$.

F  Effects of 7 days of VEGFA ± 1 μM 5′-azacytidine (5′aza) on miR-128 expression in lines indicated. PEO1R; *$P = 0.0451$, **$P < 0.0001$, ‡$P < 0.0001$, OCI-C5X; *$P = 0.0048$, **$P = 0.0003$, ‡$P = 0.0033$.

Data information: All graphed data show mean ± SEM for at least 3 different biologic experiments with at least three technical repeats within each assay. Differences between two groups were assayed by Student's *t*-test, and multiple treatment groups were compared by ANOVA. See also Fig EV4.

The ALDH1[+] population contained more sphere-forming cells than ALDH1[−] cells (Fig 7C). VEGFA treatment of sorted populations for 7 days significantly increased the % of sphere-forming cells only in the ALDH1[+] ($P = 0.0072$), not the ALDH1[−] subpopulation ($P = 0.21$; Fig 7C). Thus, prolonged VEGFA exposure increases the abundance of sphere-forming cells, but this effect is limited to an ALDH1[+] target population.

To further investigate VEGFA effects on ovarian CSCs, ALDH1-positive and ALDH1-negative cells were isolated by flow cytometry and grown with or without VEGFA over 8 days. The number and phenotype of their progeny were assayed every 2 days. The progeny of both ALDH1[+] and ALDH1[−] cells increased at similar rates in 2D culture (Fig EV6A). The growth of either ALDH1[+] or ALDH1[−] populations over the next 8 days was similar with or

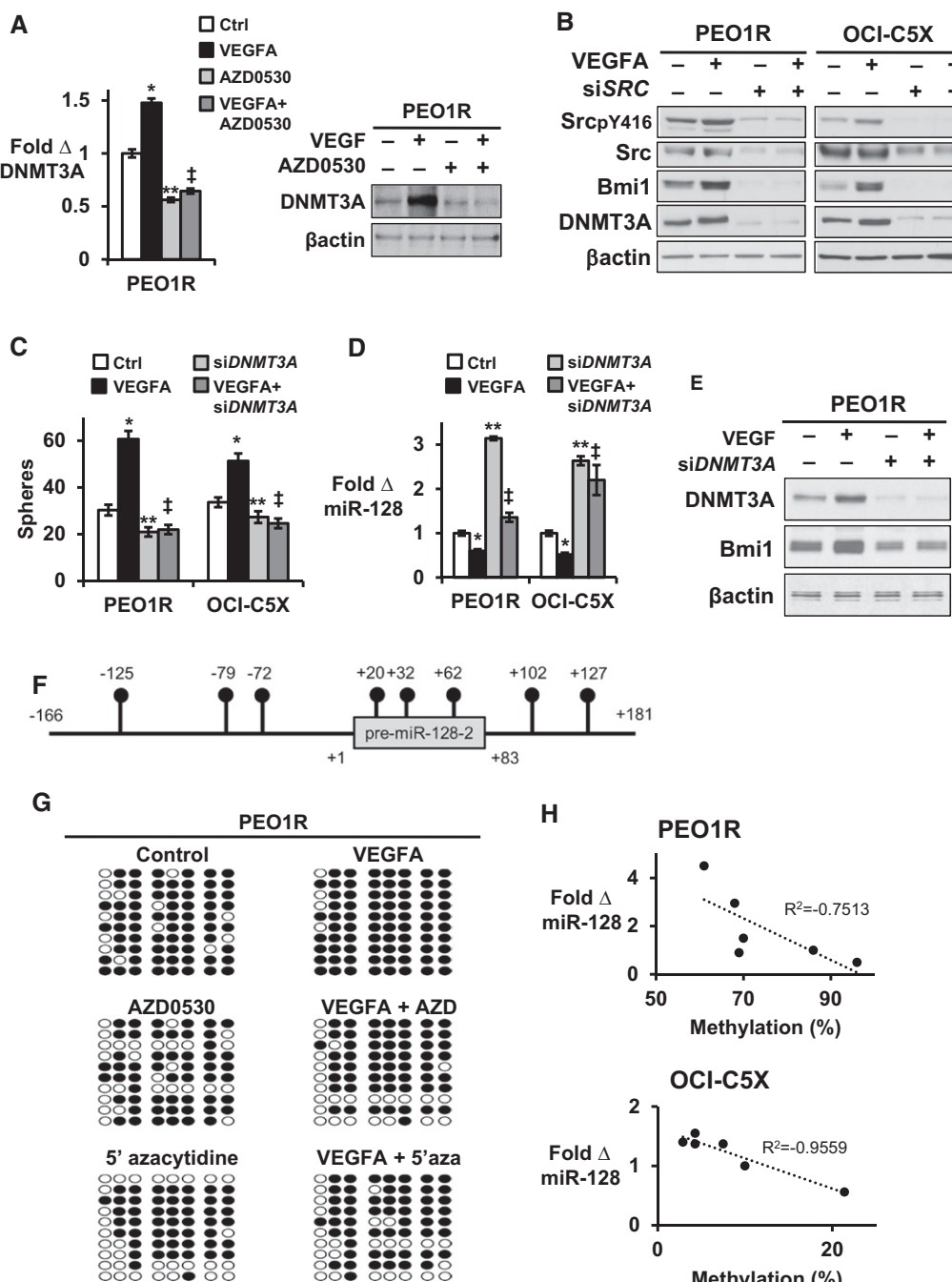

**Figure 5. VEGFA-driven sphere formation and Bmi1 upregulation are Src and DNMT3A dependent and mediated by DNA methylation of *miR-128-2*.**

A    Effects of 7 days of VEGFA on DNMT3A expression by qPCR (left) or Western blot (right) ± Src inhibition (1 μM AZD0530) over the final 48 h in PEO1R. *$P$ = 0.0031, **$P$ = 0.0051, ‡$P$ = 0.0163.

B–E  (B) PEO1R or OCI-C5X cells were transduced with siRNA to *SRC* or scrambled controls 48 h prior to VEGFA treatment for 7 days. Western blots show Src, pSrc, Bmi1, and DNMT3A levels. (C–E) Cells were transduced with either siRNA to *DNMT3A* or control siRNA (± si*DNMT3A*) for 48 h prior to 7-day treatment ± VEGFA and then (C) sphere formation was assayed for PEO1R (1,000 cells) and OCI-C5X (4,000 cells) cells (PEO1R; *$P$ = 0.0002, **$P$ = 0.0061, ‡$P$ = 0.0091, OCI-C5X; *$P$ = 0.0013, **$P$ = 0.0283, ‡$P$ = 0.0061) and (D) effects of si*DNMT3A* on VEGFA-mediated miR-128 repression were assayed by qPCR (PEO1R; *$P$ = 0.0149, **$P$ < 0.0001, ‡$P$ = 0.0023, OCI-C5X; *$P$ = 0.029, **$P$ < 0.0001, ‡$P$ = 0.0003), and (E) DNMT3A and Bmi1 levels were assayed by Western blot.

F, G  (F) *miR-128-2* genomic region with number showing the position of CpG sites from the 5′ end of pre-*miR-128-2* that were tested by bisulfite sequencing. (G) Results of bisulfite sequencing of the *miR-128-2* region in PEO1R. Open circles indicate unmethylated, and filled circles indicate methylated CpG sites.

H    Expression of miR-128 in PEO1R and OCI-C5X is inversely correlated with miR-128 methylation.

Data information: All graphed data show mean ± SEM for at least 3 different biologic experiments with at least three technical repeats within each assay. Differences between multiple treatment groups were compared by ANOVA. See also Fig EV5.

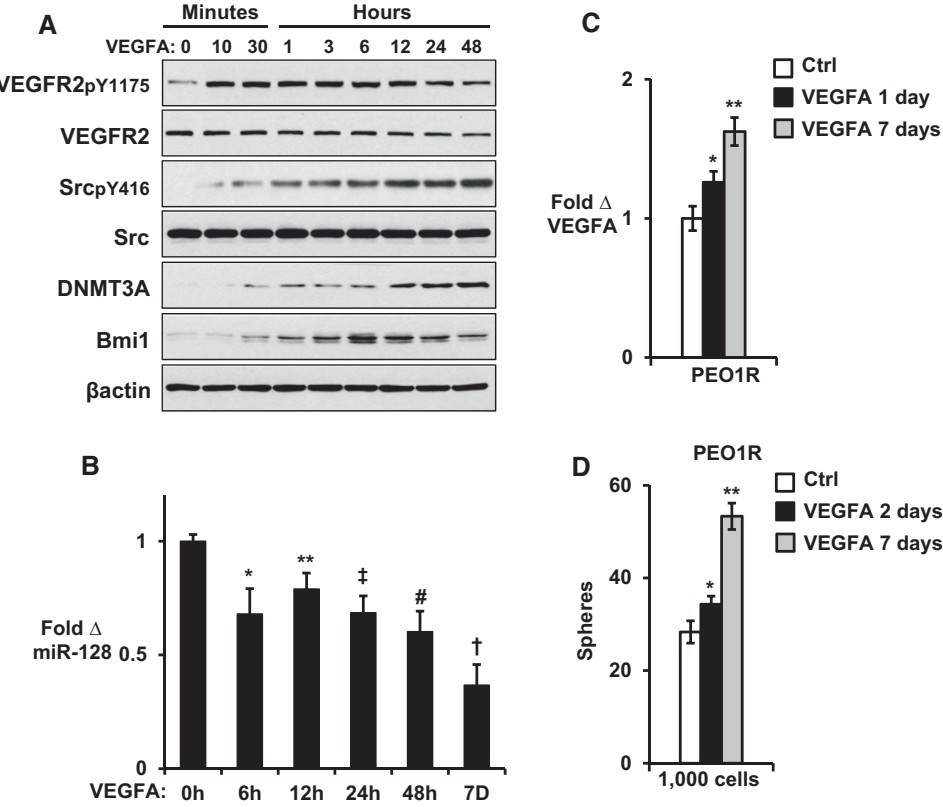

**Figure 6. Kinetics of VEGFA-mediated VEGFR2 and Src activation, DNMT3 upregulation, miR-128 decline, and increased Bmi1 and sphere formation.**

A   VEGFA effects on indicated proteins in PEO1R cells at indicated times in minutes or hours.
B   VEGFA effects on miR-128 in PEO1R cells at indicated times in hours (h) or days (D). *P = 0.0379, **P = 0.2627, ‡P = 0.0424, #P = 0.0073, †P = 0.0001.
C   VEGFA effects on VEGFA in PEO1R cells at indicated times. *P = 0.1124, **P = 0.003.
D   Indicated cells were pre-treated in 2-day culture for 2 or 7 days with 10 ng/ml VEGFA or vehicle only and then plated into sphere assays. *P = 0.2073, **P = 0.0006.

Data information: All graphed data show mean ± SEM for at least 3 different biologic experiments with at least three technical repeats within each assay. Differences between multiple treatment groups were compared by ANOVA.

without VEGFA exposure (Fig EV6B and C). Untreated ALDH1⁺ cells gave rise to ALDH1-negative cells over time, compatible with asymmetric cell division (Fig 7D). VEGFA exposure increased the number of ALDH1⁺ cells generated from ALDH1⁺ cells compared with vehicle-treated controls. The fraction of ALDH1⁺ cells remained higher and VEGFA decreased the generation of ALDH1⁻ from ALDH1⁺ precursors (Fig 7D). That VEGFA-treated ALDH1⁺ cells produced more ALDH1⁺ progeny over time (Fig 7E), is compatible with a VEGFA-induced shift from asymmetric division toward self-renewal of ALDH1⁺ ovarian CSC. Notably, sorted ALDH1⁻ cells yielded only ALDH1⁻ progeny over time (Fig EV6D). Taken together, our data are consistent with baseline activation of Src governing *DNMT3A* expression and Bmi1 in the ALDH1⁺ population. This pathway appears to be further activated following VEGFA stimulation.

## High VEGFA and decreased *miR-128-2* associate with poor OVCA outcome

High intratumor VEGFA correlates with poor outcome for various cancers, but OVCA studies have been limited by small numbers and not all show significance (Yu *et al*, 2013). A recent OVCA

meta-analysis indicated high intratumor VEGFA associates with poor outcome, but reported heterogeneity among studies and a greater prognostic import for *VEGFA* in early- than in late-stage disease (Yu *et al*, 2013). We evaluated *VEGFA* gene expression by Kaplan–Meier analysis in over one thousand primary OVCA (*n* = 1,305 for progression-free survival, PFS) using http://kmplot.com/analysis. OVCAs with reduced *VEGFA* expression had significantly better PFS, shown for the lowest quartile of *VEFGA* [HR 1.37, 95% CI (1.18–1.6), *P* = 3.9e-05] and below the median [HR 1.23, 95% CI (1.08–1.41), *P* = 0.0019] (Fig 8A).

Among four hundred and eighty five primary high-grade serous ovarian cancers (HGSOCs) in The Cancer Genome Atlas (TCGA) database with gene and miRNA expression, and outcome data available, tumors with the lowest *miR-128-2* expression associated with the worst disease-free survival (DFS, *n* = 396, *P* = 0.038) and overall survival (OS, *n* = 485, *P* = 0.023; Fig 8B). HGSOC in the top *VEGFA* expression quartile with the lowest quartile of *miR-128-2* expression showed a significantly worse disease-free survival (DFS) than those with high *miR-128-2* levels [DFS 14.8 versus 21.6 months; HR 1.98, 95% CI (1.04–3.75), *P* = 0.034] (Fig 8C), supporting the notion that a subset of aggressive OVCAs have high VEGFA-driven loss of *miR-128-2* expression.

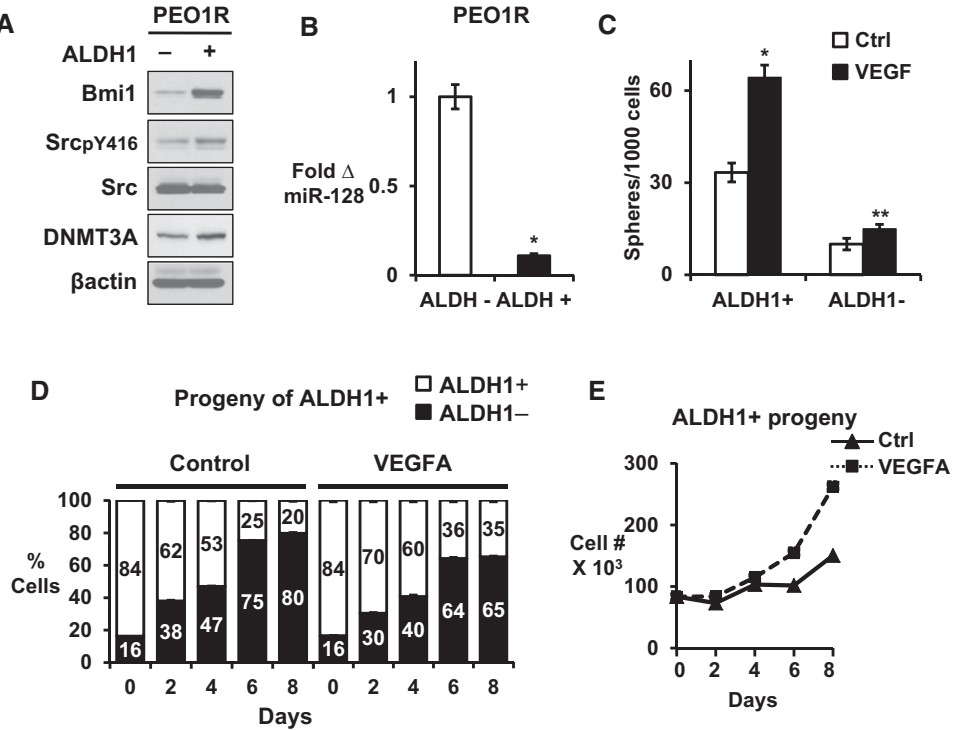

**Figure 7. Effect of VEGFA on ALDH⁺ population maintenance.**

ALDH1-positive and ALDH1-negative cells were flow sorted.

A   Western blots show Bmi1, pSrc, Src, and DNMT3A levels in sorted ALDH1⁺ and ALDH1⁻ PEO1R cells.

B   miR-128 expression in ALDH1⁺ and ALDH1⁻ PEO1R cells. *$P = 0.0037$.

C   Sorted ALDH1⁺ and ALDH1⁻ PEO1R cells were plated into sphere assays ± VEGFA. Spheres > 75 μm were counted at 14 days. *$P = 0.001$, **$P = 0.057$.

D   Sorted ALDH1⁺ and ALDH1⁻ cells were reseeded into culture. The proportion of ALDH1⁻ or ALDH1⁺ cells arising from ALDH1⁺ cells over the next 8 days in culture was serially assayed and graphed as mean % ± SEM.

E   Growth curves of ALDH1⁺ progeny arising from ALDH1⁺ cells over 8 days in culture with or without added VEGFA.

Data information: All graphed data show mean ± SEM for at least 3 different biologic experiments with at least three technical repeats within each assay. Differences between two groups were assayed by Student's *t*-test. See also Fig EV6.

## Discussion

Although increasing data suggest that a cancer stem-like cell subpopulation mediates treatment resistance and disease recurrence, the extracellular growth factors and pathways that mediate stem-like cell expansion are still too poorly defined to permit specific targeting. Present data provide novel evidence that VEGFA, produced by OVCA cells and their stroma, works through VEGFR2 to stimulate Src activation and increase CSC. Rapidly growing tumors experience central hypoxia as growth outstrips new vessel formation. VEGFA, whose expression is induced in hypoxia, would support both angiogenesis and expansion of stem-like cancer cells. While VEGFA may not be required for basal CSC maintenance in Src-activated OVCA, hypoxia would permit VEGFA to further stimulate Src and induce *DNMT3A* above basal levels required for ongoing CSC maintenance, leading to hypermethylation and silencing of *miR-128-2*, increasing Bmi1 to expand CSC. Bmi1 is part of the multiprotein PRC complex and modest increases in its levels can change complex stoichiometry with significant epigenetic consequences. VEFGA-driven Bmi1 upregulation is followed by increases in cMyc, KLF4, and Sox2, all major drivers of CSC self-renewal, consistent with observations that Bmi1 regulates ES-TFs, including Sox2 and Klf4 (Molofsky *et al*, 2005; Siddique & Saleem, 2012).

Tumor suppressor miRs are frequently extinguished in cancers. For example, miR-148a, miR-34b/c, and miR-9 downregulate oncogenes, including *c-MYC, E2F3, CDK6*, and *TGIF2* and are often reduced in cancers. 5′-azacytidine restored miRNA expression, indicating that DNA methylation repressed these tumor suppressor miRNAs to increase metastasis (Suzuki *et al*, 2012). Methylation also downregulates miR-129-2 to induce *SOX4* (Croce, 2009), and miR-143 to upregulate the *MLL-AF4* oncogene in leukemia (Dou *et al*, 2012). While many such miRNAs are methylation sensitive, the specific methyltransferases governing their expression have not been identified. Present work links a cytokine, VEGFA, with Src pathway activation and identifies DNMT3A as a novel regulator of site-specific *miR-128-2* methylation, Bmi1 upregulation, and expansion of stem-like OVCA cells. Our data are consistent with recent findings that DNA methyltransferase inhibition by SGI-110 reduced the abundance of OVCA cells with stem-like properties including ALDH1⁺ *in vitro* and tumor-initiating cells *in vivo*, and resensitized cisplatinum-resistant A2780 cells to platinum therapy (Wang *et al*, 2014). In both the PEO1R line and primary OCI-C5X culture, miR-128 expression was strongly inversely correlated with *miR-128-2*

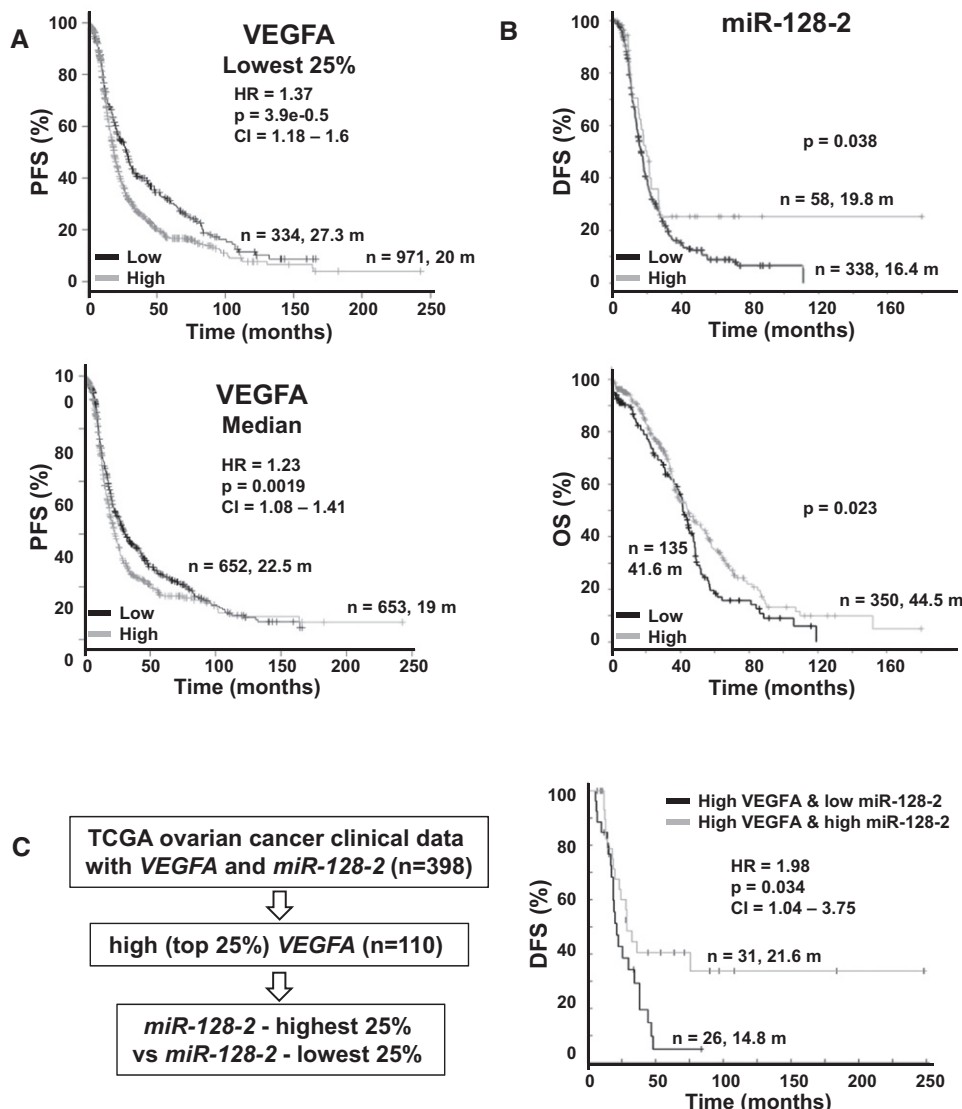

**Figure 8. Prognostic value of *VEGFA* and *miR-128-2* expression in ovarian cancer patients.**

A  Kaplan–Meier plot of progression-free survival (PFS) of OVCA patients, stratified by *VEGFA* expression of using KM Plotter (http://kmplot.com/analysis/). Patients with available clinical data: PFS, n = 1,305.

B  Kaplan–Meier plot of disease-free survival (DFS) and overall survival (OS) of TCGA high-grade serous ovarian cancer (HGSOC) patients, stratified by *miR-128-2* expression using cBioPortal (http://www.cbioportal.org/). Patients with available clinical data: DFS, n = 396; and OS, n = 485.

C  Workflow for TCGA data classification. Clinical data for HGSOC DFS were obtained from cBioportal, which are graphed by *VEGFA* and *miR-128-2* levels.

methylation ($R^2 = -0.7513$ and $R^2 = -0.9559$, respectively). Interestingly, the early-passage primary OVCA culture, OCI-C5X, showed considerably less *miR-128-2* methylation and higher *miR-128-2* expression than the extensively passaged, highly tumorigenic PEO1R line, potentially reflecting greater malignancy of the original PEO1R tumor *in vivo* or progression during culture.

Present data suggest that in contrast to hematologic malignancies where inactivating *DNMT3A* mutations reveal a tumor suppressor role (Fernandez *et al*, 2012; Yang *et al*, 2015), DNMT3A plays an oncogenic role as a key effector of VEGFA-driven ovarian CSC expansion. Epigenetic regulators, including EZH2 and DNA methyltransferases, appear to have dual roles,

acting as tumor suppressors or oncogenes in different malignancies, depending on the differentiation state of cells undergoing malignant change (Fernandez *et al*, 2012). Present work suggests that DNMT3A also plays a dual role. In addition to its tumor suppressor role in leukemia, DNMT3A also mediates stem-like cell expansion in ovarian cancers. VEGFA induced *DNMT3A*, but not related *DNMT3B* or *DNMT1*, and *DNMT3A* knockdown prevented VEGFA-mediated miR-128 loss and the increases in Bmi1 expression and OVCA tumor sphere formation *in vitro*. *BMI1* knockdown abolished VEGFA-dependent increases in sphere formation and in ovarian tumor-initiating cells *in vivo*. DNMT3A appears to act downstream of Src, which is activated in over 70% of OVCA

    

(Simpkins *et al*, 2012), and is further stimulated by VEGFA to promote tumorigenesis via *miR-128-2* methylation, increasing Bmi1 and CSC. Bmi1 thus emerges as a critical mediator of VEGFA-driven ovarian CSC expansion. Notably, Bmi1 overexpression induces chemotherapy and radiation resistance in various cancers (Siddique & Saleem, 2012) and platinum resistance in OVCA (Wang *et al*, 2011) compatible with a role for Bmi1 to drive treatment-resistant ovarian CSC.

OVCA is usually diagnosed late, and despite initial chemoresponsiveness, resistant subpopulations emerge rapidly and lead to patient demise (Winter *et al*, 2007). In contrast to other cancers, where bevacizumab has been less encouraging, several clinical trials support the use of bevacizumab with chemotherapy for advanced or recurrent ovarian cancer [for review, see Eskander and Tewari (2014)]. The GOG218 trial of 1,873 advanced OVCA patients showed bevacizumab added to carboplatinum and paclitaxel delayed disease progression (Burger *et al*, 2011). In earlier-stage OVCA, bevacizumab also prolonged the disease-free interval before progression (Perren *et al*, 2011). Two trials for recurrent OVCA showed bevacizumab also delayed disease progression in this context (Eskander & Tewari, 2014). Ongoing Phase 3 trials are evaluating other VEGFA/VEGFR-targeted therapies. Of these, the ICON6 trial of chemotherapy with or without the VEGFR inhibitor, cediranib, for platinum-sensitive recurrent OVCA is notable, since it is the first antiangiogenic trial to report improvement in both progression-free [HR 0.57, 95% CI (0.45–0.74)] and overall survival [HR0.70, 95% CI (0.51–0.99)] (Eskander & Tewari, 2014). Despite these promising findings, the median extension of the disease-free interval is less than 6 months (Eskander & Tewari, 2014) and disease progression is inevitable.

Recurrent cancers are enriched for stem-like cells. OVCA cells bearing stem cell-like markers CD44 and CD133, and ALDH1 activity are increased after chemotherapy (Shah & Landen, 2014). A recent meta-analysis showed elevated intratumor and, more significantly, serum VEGFA in OVCA both significantly associate with drug resistance and death (Yu *et al*, 2013). Our analysis of over one thousand OVCA, largely comprised of the most aggressive HGSOC subtype, shows those with the lowest intratumor *VEGFA* expression have significantly improved survival. Among HGSOCs in the highest VEGFA expression quartile, those with decreased miR-128 levels show a significantly worse outcome, supporting the existence of an aggressive HGSOC subset with VEGFA-driven miR-128 repression. There are few datasets with gene, miRNA, and clinical OVCA data available. The present analysis provides important *in vivo* human data to confirm the molecular pathway identified herein. Taken together, present findings link high VEGFA to treatment resistance and death through stimulation of ovarian CSCs.

Present work illuminates the limited efficacy of bevacizumab and other antiangiogenic/chemotherapy regimens. VEGFA blockade, by interrupting angiogenesis, creates tumor hypoxia. Hypoxia stimulates HIF-1α-dependent *VEGFA* induction (Goel & Mercurio, 2013), which would stimulate the most aggressive hypoxia-tolerant, chemoresistant CSC to reseed local and metastatic niches. Indeed, tumors surviving bevacizumab were shown to have increased CSC, due to the effects of hypoxia to upregulate VEGFA (Conley *et al*, 2012). VEGFA can act via VEGFR2 (Zhao *et al*, 2014) and via the neuropilin receptor (Goel & Mercurio, 2013) to drive VEGFA induction and CSC expansion in breast models. The present study reveals a novel rationale and a potential strategy for targeting VEGFA pathways in OVCA. This work used high-grade serous and clear cell OVCA models. Since high-grade serous and clear cell cancers appear to represent two distinct cancer types, arising from different sites (Crum *et al*, 2007), with different genetic drivers (Wiegand *et al*, 2010; Ince *et al*, 2015), our findings may have broader relevance, beyond OVCA, for other VEGF-driven epithelial cancers. While bevacizumab generates a resistance cycle, combined use of drugs that inhibit VEGFA signaling at multiple levels, such as inhibitors of VEGFR, Src and/or DNA methyltransferases, might hold greater promise as stem-like cell-directed treatments.

# Materials and Methods

### Cell lines and culture

PEO1R (Langdon *et al*, 1994) and OVCAR8 ovarian cancer lines were cultured as in Simpkins *et al* (2012). The OCI-C5X culture was derived from a primary clear cell ovarian cancer and cultured in OCMI medium as described (Ince *et al*, 2015) and used at passages < 40. OCI-C5X cells and OCMI medium were from the Sylvester Comprehensive Cancer Center Live Tumor Culture Core (LTCC) at UM Miller School of Medicine (contact email LTCC@med.miami.edu). Viable cell numbers were counted every 2 days over 8 days.

### Sphere formation and ALDEFLUOR assays

Spheres were seeded with or without prior 7-day VEGFA exposure (R&D) 10 ng/ml using limiting dilutions as in Ginestier *et al* (2007). Spheres > 75 μm were counted after 14–21 days. VEGFA was renewed every 48 h over 7 days. Where indicated, bevacizumab (100 μg/ml, Genentech, CA, USA) or the 2C3 VEGFR2-blocking antibody (15 μg/ml, provided by R. Brekken, UT Southwestern USA) was added immediately before VEGFA and renewed every 2 days. Where indicated, the Src inhibitor saracatinib (AstraZeneca) 1 μM was added for the final 48 h of the 7 days of VEGFA pre-treatment. ALDEFLUOR assays were as in Ginestier *et al* (2007).

### Cell cycle distribution assays and annexin V staining

Control cell cycle assays as in Zhao *et al* (2014) were done immediately prior to plating of cells into sphere assays or injection into nude mice for limiting dilution stem-like cell assays. Since saracatinib 1 μM (AZD0530) caused partial G1 arrest, all groups were subjected to a 2-day washout without drug or cytokine prior to sphere assays. Cells returned to asynchronous cycling after a 2-day saracatinib washout. Cell cycle distribution was assayed after 7 days of VEGFA followed by 2 days without cytokine (VEGFA), or following 7 days of VEGFA with AZD0530 added for 48 h (days 6 and 7) followed by 2-day washout without cytokine or AZD0530 (AZD0530 + washout). si*BMI1* cells were transfected with si*BMI1* 48 h prior to VEGFA treatment for 7 days and followed by 2 days without cytokine. All cells were proliferating asynchronously at plating. After VEGFA treatment for 7 days, cells were analyzed by FITC Annexin V Apoptosis Detection Kit I (BD Biosciences) as per manufacturer's guidelines.

## Flow sorting of ALDH1⁺ or ALDH1⁻ cells

Cells were stained with ALDEFLUOR™ (Stem Cell Technologies, Durham, NC) and then ALDH1⁺ or ALDH1⁻ cells separated by FACSAria II (BD Biosciences). Lineage tracing with MitoTracker dyes (Thermo Fisher Scientific and Invitrogen) was not possible because these dyes all caused cytotoxicity.

## Tumor-initiating stem-like cell assays

The University of Miami (UM) Animal Care and Use Committee approved all animal work. For limiting dilution xenograft assays, PEO1R was transduced or not with si*BMI1* or scrambled siRNA (Santa Cruz Biotech) controls for 48 h and then treated with ± VEGFA 10 ng/ml for 7 days. One hundred and twenty-four 5-week-old, female NOD/SCID mice were injected with cells from four experimental groups (untreated scrambled siRNA controls, VEGFA-treated scrambled siRNA controls, VEGFA-treated si*BMI1*-transduced, and si*BMI1*-transduced). Animals were injected in the mammary fat pad with 100 (10 mice/group), 1,000 (eight mice/group), 10,000 (eight mice/group), or 100,000 (five mice/group) cells in 100 µl Matrigel (BD Biosciences) as in Ginestier *et al* (2007). Tumors were measured twice per week. Mice without tumors were followed at least 9 months as per UM Animal Care and Use Committee standards.

## miRNA RT–PCR, siRNA, antagomiR-128, and Western blot analysis

miRNA isolation was carried out using miRNeasy (Qiagen) in cells treated with or without prior VEGFA, saracatinib, or 5′-azacytidine (Sigma-Aldrich). cDNA synthesis used Ncode miR First Strand cDNA synthesis kit (Invitrogen). qPCR of miR-128 used miR-128 forward: 5′-TCACAGTGAACCGGTCTCTTT-3′ and Universal reverse primer (Ncode miR First Strand cDNA synthesis kit, Invitrogen). Three different siRNA oligos to each of SRC, BMI1, and DNMT3A and scrambled controls were purchased from Santa Cruz Biotech (Dallas). Knockdown was assayed after 48 h by Western blot as described (Zhao *et al*, 2014). AntagomiR-128 and its control were purchased from Exiqon (Denmark) and transduced into cells as per manufacturer's protocol. Bmi1, pStat3, Myc, Klf4, Oct4, SrcpY416, Src, VEGFR2pY1175, VEGFR2, and DNMT3A antibodies were from Cell Signaling. β-actin antibody was from Santa Cruz Biotech. Densitometric analysis was carried out for Western blot data using three different exposures of three different biologic assays and data presented as fold change expression ± SEM.

## Luciferase assays

PEO1R, OVCAR8, and OCI-C5X cells were treated with 10 ng/ml VEGFA for 4 days prior to transfection with 500 ng pEZX-MT01-*BMI1*-3′UTR plasmid (GeneCopoeia) followed by 72-h further VEGFA treatment. Firefly and Renilla luciferase reporter activity was measured using Luc-Pair™ Duo-Luciferase Assay Kit 2.0 (Gene-Copoeia) as per manufacturer's instructions.

## Bisulfite sequencing

Bisulfite conversion was performed with Epitect Fast DNA Bisulfite Kit (Qiagen) using genomic DNA from the VEGFA, saracatinib, or 5′-azacytidine-treated PEO1R and OCI-C5X cell lines. Primers (forward: TGTTTTTAAGGTTAGGGAATTAAATTAG, reverse: TCAACA AAAATAACACAAACCTCTC) amplified *miR-128-2* in bisulfite-converted genomic DNA. PCR products were gel-purified and cloned with pcDNA™3.1/V5-His TOPO® Expression kit (Life Technologies). DNA from bacterial colonies was subjected to bisulfite sequencing.

## Survival analysis

Publicly available ovarian cancer datasets from the National Center for Biotechnology Information (GSE14764, GSE9891, GSE51373, GSE3149, GSE30161, GSE27651, GSE26712, GSE26193, GSE23554, GSE19828, GSE18520, GSE15622) and the ovarian cancer dataset from The Cancer Genome Atlas (TCGA; using cBioPortal [http://www.cbioportal.org/], downloaded 4-10-15) were analyzed using KM plotter (http://kmplot.com/analysis/) to create Kaplan–Meier (KM) curves for PFS and OS versus *VEGFA* expression (Gyorffy *et al*, 2012; Mihaly *et al*, 2013). The KM curves for disease-free survival and OS with *miR-128-2* expression used TCGA data. OVCAs with high *VEGFA* and high or low *miR-128-2* expression from TCGA were compared by cBioPortal to create Kaplan–Meier curves.

## Statistical analysis

For *in vitro* work, data are graphed from ≥ 3 biologic experiments as means ± standard error of the mean (SEM). Comparisons of > two groups was made using one-way analysis of variance (ANOVA) followed by Student–Newman–Keul's *post hoc* analysis or two-way ANOVA followed by Bonferroni *post hoc* tests. Comparisons of two groups was made using Student's *t*-test. Tumor-initiating cell frequency was calculated by L-Calc Limiting Dilution Software (http://www.stemcell.com/en/Products/All-Products/LCalc-Softwa re.aspx) from STEMCELL Technologies.

## Study approval

All animal work was UM SOM IACUC-approved. All analyses of human OVCA for VEGFA or miRNA-128 were based on *in silico* analysis of retrospective datasets including National Center for Biotechnology Information and the OVCA TCGA dataset using cBioPortal. This work is not considered "Human Subjects Research".

**Expanded View** for this article is available online.

## Acknowledgements

We thank Marcus Peter and Brian Slomovitz for helpful discussions of the work. This work was supported by a grant from the Breast Cancer Research Foundation to JMS.

## Author contributions

KJ and MK designed and performed experiments, analyzed data, and wrote the paper. CG and FS designed and/or performed experiments and analyzed data. KJ and MK performed statistical data analysis. TAI provided primary ovarian cancer cultures and methodology for their culture and assisted with experimental design. JMS designed experiments, obtained funding to support the work, analyzed data, and wrote the paper.

## The paper explained

### Problem

VEGFA promotes tumor angiogenesis and growth. While VEGFA-targeted therapy can prolong time to disease recurrence and death in advanced OVCA, treatment resistance invariably emerges. Ovarian cancer is usually diagnosed late and, despite initial chemoresponsiveness, resistant subpopulations emerge rapidly and lead to patient demise. Chemotherapy-resistant populations have been shown to be enriched in tumor-initiating stem-like cells. Recent work has implicated VEGFA as a driver of malignant stem-like cells in certain cancers, but the relevance to ovarian cancer and the downstream pathways through which VEGFA-driven stem cell expansion were not known.

### Results

Here, we show in both high-grade serous OVCA lines and a primary clear cell OVCA culture that VEGFA drives ovarian cancer stem-like cell expansion via Src-dependent upregulation of DNMT3A, leading to methylation-dependent loss of *miR-128-2* and Bmi1 upregulation. Analysis of among over 1,300 human ovarian cancers showed those with elevated VEGFA together with low *miR-128-2* have the worst outcome, confirming the existence of an aggressive subset in which VEGFA is linked to *miR-128-2* loss.

### Impact

This work identifies DNMT3A as a novel pro-oncogenic downregulator of *miR-128-2* biogenesis and illuminates the limited efficacy of bevacizumab and other antiangiogenic/chemotherapy regimens. VEGFA blockade, by interrupting angiogenesis, creates tumor hypoxia, which would stimulate HIF-1α-dependent *VEGFA* induction to stimulate hypoxia-tolerant, chemoresistant stem-like cells to reseed local and metastatic niches. Present work provides a novel rationale to support preclinical and clinical efforts to block VEGFA signaling at multiple levels, using inhibitors of VEGFR, Src and DNA methyltransferases to target ovarian cancer stem-like cells.

## Conflict of interest

The authors declare that they have no conflict of interest.

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
