## [Review Process File · EMBO Molecular Medicine]

VEGFA activates an epigenetic pathway upregulating ovarian cancer initiating cells

Kibeom Jang, Minsoon Kim, Candace Gilbert, Fiona Simpkins, Tan A Ince, Joyce M Slingerland

Corresponding author: Joyce Slingerland, University of Miami Miller School of Medicine

Review timeline:

Submission date:	19 July 2016
Editorial Decision:	19 August 2016
Revision received:	21 November 2016
Editorial Decision:	09 December 2016
Revision received:	16 December 2016
Accepted:	19 December 2016

Transaction Report:

Editor: Roberto Buccione

1st Editorial Decision

19 August 2016

Thank you for the submission of your manuscript to EMBO Molecular Medicine. We are very sorry that it has taken longer than usual to get back to you on your manuscript.

As I had anticipated, due to the holiday season we experienced unusual difficulties in securing three willing and appropriate reviewers. As a further delay cannot be justified I have decided to proceed based on the two available evaluations.

Although #1 is more reserved, both reviewers are largely positive but raise a few of issues that require your action. I will not go into detail, as their comments are quite clear.

Reviewer 1 feels that a mechanistic analysis upgrade is required to strengthen the findings and increase their impact. We agree, especially considered the novelty issue mentioned by reviewer 2. I should also mention that, notwithstanding the overall interest of your findings, we shared the same concern while initially deciding on your manuscript. Reviewer 2 also raises a very interesting general point concerning the cellular origin of ovarian cancer(s) and suggests that your manuscript be reworded to reflect that it is mostly focused on the HGSC type.

In conclusion, while publication of the paper cannot be considered at this stage, we would be pleased to consider a revised submission, with the understanding that the Reviewers' concerns must be addressed including with additional experimental data where appropriate and that acceptance of the manuscript will entail a second round of review.

Please note that it is EMBO Molecular Medicine policy to allow a single round of revision only and that, therefore, acceptance or rejection of the manuscript will depend on the completeness of your

responses included in the next, final version of the manuscript.

As you might know, EMBO Molecular Medicine has a "scooping protection" policy, whereby similar findings that are published by others during review or revision are not a criterion for rejection. However, I do ask you to get in touch with us after three months if you have not completed your revision, to update us on the status. Please also contact us as soon as possible if similar work is published elsewhere.

Please note that EMBO Molecular Medicine now requires a complete author checklist (<http://embomolmed.embopress.org/authorguide#editorial3>) to be submitted with all revised manuscripts. Provision of the author checklist is mandatory at revision stage; The checklist is designed to enhance and standardize reporting of key information in research papers and to support reanalysis and repetition of experiments by the community. The list covers key information for figure panels and captions and focuses on statistics, the reporting of reagents, animal models and human subject-derived data, as well as guidance to optimise data accessibility. The Author checklist will be published alongside the paper, in case of acceptance, within the transparent review process file.

Finally, we now mandate that all corresponding authors list an ORCID digital identifier. You may do so through our web platform upon submission and the procedure takes <90 seconds to complete. We also encourage co-authors to supply an ORCID identifier, which will be linked to their name for unambiguous name identification.

I look forward to seeing a revised form of your manuscript as soon as possible.

***** Reviewer's comments *****

Referee #1 (Remarks):

In this manuscript the authors show that VEGF favours the expansion of aldehyde dehydrogenase positive ovarian cancer stem cells. The authors define a mechanisms connecting VEGFR2 with src, which in turn positively regulates the expression of Bmi-1. The effect of VEGF on Bmi1 expression is post-transcriptional. Actually VEGF promotes the methylation of miR-128 resulting in its repression. miR-128 is known to negatively regulate Bmi1 expression. Interestingly, high levels of VEGF with loss of miR128 select a subpopulation of ovarian cancer with a poor prognosis. The paper contains relevant information but the present version requires new experiments to explain better the mechanisms involved and some controls.

CRITICISMS

The effect of VEGF on the expansion of CSF is detectable after 7 days of treatment and the authors show that the expression of phosphorylated src and Bmi1 started after 6 hours and persisted up to day 7. This prolonged stimulation of src requires a deeper analysis than that proposed by the present version of the manuscript.

Do the cells undergo a VEGF autocrine pathway? Does this putative mechanism contribute in the prolonged src activation? Which is the minimal time required to VEGF to trigger src activation, increased expression of Bmi1 and DNMT3A. Actually the authors stimulate spheres with VEGFA every days. Which is the time course of src activation, VEGFR2 phosphorylation, Bmi1 and DNMT3A expression in the 2 days frame?

In the experimental conditions described by the authors, which are the VEGFR2 phosphorylation sites crucial for the described activities?

The authors show that VEGFA does not modify the cell cycle in the sphere? Does it has an anti-apoptotic effect? Does the lack of effect on cell cycle occur in 2D culture conditions?

The role of src in the VEGF-mediated expansion of CSF is based on the use of a specific inhibitor (Fig 2). In other experiments the (Fig 6) the authors use a specific siRNA. I suggest planning an experiment on sphere formation by showing the effect of src silencing.

The authors demonstrate the pivotal role of miR128. Does antagomir128 mimic the effect of

VEGFA?

Referee #2 (Comments on Novelty/Model System):

The paper uses accepted methodology for studying cancer stem cells, CSCs therefore I conclude that the model systems are adequate and the technical quality high. I would rate the novelty as medium as previous papers have shown that VEGF stimulates CSCs. The medical impact is probably high as there are supporting data from large numbers of patient samples and an effect on CSC expansion may in part explain the brevity of responses to anti-VEGF therapies.

Referee #2 (Remarks):

This paper investigates the molecular mechanisms by which VEGF stimulates expansion of cancer stem cells, CSCs in ovarian cancer cell lines and relates the data to ovarian cancer patient data. In general the work is carefully performed but the manuscript requires revision to reflect our current knowledge of the human cancers collectively termed 'ovarian'. We now know that invasive 'ovarian' cancers are actually four different cancers, none of which are thought to arise in the ovary, high grade serous, HGSC, (fallopian tube origin) endometrioid and clear cell (arise in the endometrium) mucinous (probably a gut metastasis). In this paper PEO1 cells are HGSC I believe as are the OC1-C5X cells. Domcke et al in a Nature Communications paper in 2013 also described OVCAR8 as 'possibly HGSC' and they do have a TP53 mutation which is characteristic of HGSC. Therefore I suggest that to reflect our current knowledge OVCA should be replaced with HGSC and all patient data used should be from HGSC patients. The other 'ovarian' cancers have different genetic drivers and it is possible that their CSCs may have different markers and different responses to growth factors.

Minor point

Page 20 Methods - is 'mammospheres' the correct term

1st Revision - authors' response

21 November 2016

REFEREE 1.

1. *The effect of VEGF on the expansion of CSF is detectable after 7 days of treatment and the authors show that the expression of phosphorylated Src and Bmi1 started after 6 hours and persisted up to day 7. This prolonged stimulation of Src requires a deeper analysis than that proposed by the present version of the manuscript.*

Do the cells undergo a VEGF autocrine pathway? Does this putative mechanism contribute in the prolonged Src activation? Which is the minimal time required to VEGF to trigger Src activation, increased expression of Bmi1 and DNMT3A. Actually the authors stimulate spheres with VEGFA every days. Which is the time course of Src activation, VEGFR2 phosphorylation, Bmi1 and DNMT3A expression in the 2 days frame?

First, we would like to correct the referee's statement that "the authors stimulate spheres with VEGFA every days." We treated cells in 2D culture with VEGFA for a 7 day period with 10ng/ml VEGFA, which was replenished every 2 days prior to plating cells for sphere formation assay. Cells received no further VEGFA during the 2 week interval of sphere formation after seeding onto ultralow adhesion plates.

As requested by Referee 1, we present a more detailed kinetic time course in new data in Fig 6 showing that short term VEGFA exposure rapidly activates VEGFR2pY1175 leading to increased SrcpY416. Both DNMT3 and BMI1 increased gradually within the first 6 hrs after VEGFA addition. Notably, miR128 levels fell significantly within the first 6-12 hrs, and the decline was sustained and progressive over the next 7 days. Thus, while VEGFR2 and Src signaling is triggered within minutes, the signaling increases progressively and the epigenetic consequences of Src-dependent DNMT3A upregulation, and miR128 loss were felt progressively over the successive 3-4 DNA replication cycles over 7 days. In keeping with an epigenetic effect and not merely a signaling dependent mechanism, the increase in sphere forming ability was not fully felt until after several cell

divisions: VEGFA exposure for 48 hrs was not sufficient to increase spheres and a 7 day exposure was required. These data support that long term VEGFA exposure is required increase the abundance of CSC. VEGFA treatment does indeed upregulate *VEGFA* gene expression, which would leading to a VEGFA-driven feed-forward mechanism to sustain and amplify Src activation with prolonged exposure.

2. *In the experimental conditions described by the authors, which are the VEGFR2 phosphorylation sites crucial for the described activities?*

We assayed VEGFR2 activity using an antibody reactive to VEGFR2pTyr1175, a major phosphorylation site required for receptor activation (See Fig 6A).

The authors show that VEGFA does not modify the cell cycle in the sphere? Does it have an anti-apoptotic effect? Does the lack of effect on cell cycle occur in 2D culture conditions?

In ExtraView Fig 1A we showed VEGFA exposure did not change the cell cycle in 2D culture conditions, nor did it affect cell cycle distribution when cells were grown as spheres for 48 or 7 days (48 hr exposure cell cycle effects, Extra View Fig 1A). The short term increase in cell number over 7 days was not affected by cell death (ExtraView Fig 1C). We have added new data to show that Annexin V staining for apoptotic cells was unchanged by VEGFA exposure (see ExtraView Fig 1B). Notably VEGFA also did not change cell viability after 7 days (ExtraView Fig 3B).

3. *The role of Src in the VEGF-mediated expansion of CSF is based on the use of a specific inhibitor (Fig 2). In other experiments the (Fig 6) the authors use a specific siRNA. I suggest planning an experiment on sphere formation by showing the effect of src silencing.*

The referee correctly points out that AZD0530 is a pan-Src family kinase inhibitor. As requested by Referee 1, we present new data in the revised Figure 4E showing that siRNA of SRC prevented VEGFA mediated increase in sphere formation.

4. *The authors demonstrate the pivotal role of miR128. Does antagomir128 mimic the effect of VEGFA?*

As requested by Referee 1, we present new data in the revised Figure 4E showing that antagomiR-128 decreased miRNA128 levels by QPCR and also stimulated sphere formation, even without added VEGFA. In addition, While SiRNA Src prevented the upregulation of spheres by VEGFA, siRNA Src could be rescued by miR-128 antagomir, confirming that miR-128 is regulated downstream of Src.

REFEREE 2.

Minor point

1. *Page 20 Methods - is 'mammospheres' the correct term*

This term has been changed to tumorsphere throughout

2. *This paper investigates the molecular mechanisms by which VEGF stimulates expansion of cancer stem cells, CSCs in ovarian cancer cell lines and relates the data to ovarian cancer patient data. In general the work is carefully performed but the manuscript requires revision to reflect our current knowledge of the human cancers collectively termed 'ovarian'. We now know that invasive 'ovarian' cancers are actually four different cancers, none of which are*

thought to arise in the ovary, high grade serous, HGSC, (fallopian tube origin) endometrioid and clear cell (arise in the endometrium) mucinous (probably a gut metastasis). In this paper PEO1 cells are HGSC I believe as are the OCI-C5X cells. Domcke et al in a Nature Communications paper in 2013 also described OVCAR8 as 'possibly HGSC' and they do have a TP53 mutation which is characteristic of HGSC. Therefore I suggest that to reflect our current knowledge OVCA should be replaced with HGSC and all patient data used should be from HGSC patients. The other 'ovarian' cancers have different genetic drivers and it is possible that their CSCs may have different markers and different responses to growth factors.

While PEO1R and OVCAR8 are of high grade serous ovarian cancer (HGSC) origin, the primary ovarian cancer culture, OCI-C5X, was derived from a high grade Clear Cell cancer. All of the mechanistic data in the paper were carried out with not only the HGSC cell lines but confirmed in primary Clear cell cOVCA-derived OCI-C5X. Thus we respectfully submit that our findings of a VEGFA-dependent stem cell expansion pathway induced may be more broadly applicable, and not just apply to HGSC.

Because the critiques were straightforward and the new data address their concerns simply, we ask that you consider making an editorial decision in favor of accepting the manuscript. We thank you for considering this work and look forward to hearing from you.

2nd Editorial Decision

09 December 2016

Thank you for the submission of your revised manuscript to EMBO Molecular Medicine. We have now received the enclosed reports from the referees that were asked to re-assess it. As you will see the reviewers are now globally supportive and I am pleased to inform you that we will be able to accept your manuscript pending the following final amendments:

- 1) Reviewer 2 notes that the data in the paper suggest that VEGF can have effects on cancer initiating cells from multiple cancer types. S/he therefore would like you to mention this in the discussion and to better explain the different origins of the cell lines in the results section.
- 2) Please provide individual files for figures
- 3) The manuscript must include a statement in the Materials and Methods identifying the institutional and/or licensing committee approving the experiments, including any relevant details (like how many animals were used, of which gender, at what age, which strains, if genetically modified, on which background, housing details, etc). We encourage authors to follow the ARRIVE guidelines for reporting studies involving animals. Please see the EQUATOR website for details: <http://www.equator-network.org/reporting-guidelines/improving-bioscience-research-reporting-the-arrive-guidelines-for-reporting-animal-research/>. Please make sure that ALL the above details are reported.
- 4) The nature of every author's contribution must be specified in the manuscript under the heading "Author Contributions".
- 5) We encourage the publication of source data, particularly for electrophoretic gels and blots, with the aim of making primary data more accessible and transparent to the reader. Would you be willing to provide a PDF file per figure that contains the original, uncropped and unprocessed scans of all or at least the key gels used in the manuscript? The PDF files should be labeled with the appropriate figure/panel number, and should have molecular weight markers; further annotation may be useful but is not essential. The PDF files will be published online with the article as supplementary "Source Data" files. If you have any questions regarding this just contact me.
- 6) Every published paper includes a 'Synopsis' to further enhance discoverability. Synopses are displayed on the journal webpage and are freely accessible to all readers. They include a short standfirst as well as 2-5 one sentence bullet points that summarise the paper. Please provide the synopsis including the short list of bullet points that summarise the key NEW findings. The bullet points should be designed to be complementary to the abstract - i.e. not repeat the same text. We encourage inclusion of key acronyms and quantitative information. Please use the passive voice.

Please attach this information in a separate file or send them by email, we will incorporate it accordingly. You are also welcome to suggest a striking image or visual abstract to illustrate your article. If you do please provide a jpeg file 550 px-wide x 400-px high.

Please submit your revised manuscript within two weeks. I look forward to seeing a revised form of your manuscript as soon as possible.

***** Reviewer's comments *****

Referee #1 (Remarks):

The authors did a whole revision

Referee #2 (Comments on Novelty/Model System):

The authors have used two cell lines of high grade serous ovarian cancer and one cell line of clear cell cancer origin. All information we have now shows that these are completely different cancers with different sites of origin and different genetic drivers. The data in the paper therefore suggest that VEGF can have effects on cancer initiating cells from at least two different cancer types. I think this should be mentioned in the discussion and the different origins of the cell lines made more clear in the results section.

2nd Revision - authors' response

16 December 2016

We submit now a version with the minor revisions requested by Reviewer 2.

1. As requested, the different origins of ovarian cancer lines used in the paper are made clearer in the revised results section. We also make the comment in the last paragraph of the discussion that since ovarian cancers of clear cell and high grade serous origin are very different, our data may have application to other epithelial cancers of different tissue origin also. The revisions are tracked in the word version uploaded.
2. We have uploaded individual files for each figure.
3. We have provided in the Materials and Methods the institutional committee approval of the animal experiments, and have included how many animals were used, of which gender, at what age, which strain. The animals were not genetically modified.
4. The nature of every author's contribution is specified in the manuscript under the heading "Author Contributions".
5. Given the approach of the holidays I do not think we can provide source data in a timely manner and request forgiveness for this item.
6. The Synopsis has been uploaded together with a Visual Abstract as a single file.
7. The complete author checklist was submitted with the previous revision and is uploaded on the journal website.
8. The conflict of interest statement is in the manuscript.
9. The Corresponding author ORCID digital identifier is 0000-0003-1487-8554.

Corresponding Author Name: Joyce M. Slingerland

Journal Submitted to: EMBO molecular medicine

Manuscript Number: EMM-2016-06840-V2